# LEARNING A META-SOLVER FOR SYNTAX-GUIDED PROGRAM SYNTHESIS

**Xujie Si[*1], Yuan Yang[*2], Hanjun Dai[2], Mayur Naik[1] & Le Song[2]**
[1]University of Pennsylvania, [2]Georgia Institute of Technology
[1]{xsi,mhnaik}@cis.upenn.edu
[2]{yyang754,hanjundai}@gatech.edu, lsong@cc.gatech.edu

## ABSTRACT

A general formulation of program synthesis called *syntax-guided synthesis* (SyGuS) seeks to synthesize a program that follows a given grammar and satisfies a given logical specification. Both the logical specification and the grammar have complex structures and can vary from task to task, posing significant challenges for learning across different tasks. Moreover, supervision is often unavailable for domain-specific synthesis tasks. To address these challenges, we propose a meta-learning framework that learns a transferable policy using only weak supervision. Our framework consists of three components: 1) an encoder, which embeds both the logical specification and grammar at the same time using a graph neural network; 2) a grammar adaptive policy network which enables learning a transferable policy; and 3) a reinforcement learning algorithm that jointly trains the embedding and adaptive policy with sparse reward. We evaluate the framework on 214 cryptographic circuit synthesis tasks. It solves 141 of them in the out-of-box solver setting, significantly outperforming a similar search-based approach but without learning, which solves only 31. The result is comparable to two state-of-the-art classical synthesis engines, which solve 129 and 153 respectively. In the meta-solver setting, the framework can efficiently adapt to unseen tasks and achieves speedup ranging from $2\times$ up to $100\times$.

## 1 INTRODUCTION

Program synthesis concerns automatically generating a program that satisfies desired functional requirements. Promising results have been demonstrated by applying this approach to problems in diverse domains, such as spreadsheet data manipulation for end-users (Gulwani et al., 2012), intelligent tutoring for students (Singh et al., 2013), and code auto-completion for programmers (Feng et al., 2017), among many others.

In a common formulation posed by Alur et al. (2013) called *syntax-guided synthesis* (SyGuS), the program synthesizer takes as input a logical formula $\phi$ and a grammar $G$, and produces as output a program in $G$ that satisfies $\phi$. In this formulation, $\phi$ constitutes a semantic specification that describes the desired functional requirements, and $G$ is a syntactic specification that constrains the space of possible programs.

The SyGuS formulation has been targeted by a variety of program synthesizers based on discrete techniques such as constraint solving (Reynolds et al., 2015), enumerative search (Alur et al., 2017b), and stochastic search (Schkufza et al., 2013). A key limitation of these synthesizers is that they do not bias their search towards likely programs. This in turn hinders their efficiency and limits the kinds of programs they are able to synthesize.

It is well known that likely programs have predictable patterns (Hindle et al., 2012; Allamanis et al., 2018a). As a result, recent works have leveraged neural networks for program synthesis. However, they are limited in two aspects. First, they do not target general SyGuS tasks; more specifically:

---

*The first two authors contributed equally to this work.

- They assume a fixed grammar (i.e., syntactic specification $G$) across tasks. For example, Si et al. (2018) learn loop invariants for program verification, but the grammar of loop invariants is fixed across different programs to be verified.
- The functional requirements (i.e., semantic specification $\phi$) are omitted, in applications that concern identifying semantically similar programs (Piech et al., 2015; Allamanis et al., 2017; 2018b; Dai et al., 2018), or presumed to be input-output examples (Parisotto et al., 2016; Balog et al., 2017; Devlin et al., 2017; Bunel et al., 2018; Chen et al., 2018; Vijayakumar et al., 2018; Shin et al.; Sun et al., 2018; Pu et al., 2018).

In contrast, the SyGuS formulation allows the grammar $G$ to vary across tasks, thereby affording flexibility to enforce different syntactic requirements in each task. It also allows to specify functional requirements in a manner more general than input-output examples, by allowing the semantic specification $\phi$ to be a logical formula (e.g., $f(x) = 2x$ instead of $f(1) = 2 \wedge f(3) = 6$). As a result, the general SyGuS setting necessitates the ability to capture common patterns across different specifications and grammars. A second limitation of existing approaches is that they rely on strong supervision on the generated program (Parisotto et al., 2016; Balog et al., 2017; Bunel et al., 2018). However, in SyGuS tasks, ground truth programs $f$ are not readily available; instead, a checker is provided that verifies whether $f$ satisfies $\phi$.

In this paper, we propose a framework that is general in that it makes few assumptions on specific grammars or constraints, and has meta-learning capability that can be utilized in solving unseen tasks more efficiently. The key contributions we make are (1) a joint graph representation of both syntactic and semantic constraints in each task that is learned by a graph neural network model; (2) a grammar adaptive policy network that generalizes across different grammars and guides the search for the desired program; and (3) a reinforcement learning training method that enables learning transferable representation and policy with weak supervision.

We demonstrate our meta-learning framework on a challenging and practical instance of the SyGuS problem that concerns synthesizing cryptographic circuits that are provably free of side-channel attacks (Eldib et al., 2016). In our experiments, we first compare the framework in an out-of-box solver setting against a similar search-based approach and two state-of-the-art classical solvers developed in the formal methods community. Then we demonstrate its capability as a meta-solver that can efficiently adapt to unseen tasks, and compare it to the out-of-box version.

## 2    PROBLEM FORMULATION

The Syntax-Guided Synthesis (SyGuS) problem is to synthesize a function $f$ that satisfies two kinds of constraints:

- a syntactic constraint specified by a context-free grammar (CFG) $G$, and
- a semantic constraint specified by a formula $\phi$ built from symbols in a background theory $T$ along with $f$.

One example of the SyGuS problem is cryptographic circuit synthesis (Eldib et al., 2016). The goal is to synthesize a side-channel free cryptographic circuit by following the given CFG (syntactic constraint) while ensuring that the synthesized circuit is equivalent to the original circuit (semantic constraint). In this example, the grammar is designed to avoid side-channel attacks, whereas the original circuit is created only for functional correctness and thus is vulnerable to such attacks. We henceforth use this problem as an illustrative example but note that our proposed method is not limited to this specific SyGuS problem.

We investigate how to efficiently synthesize the function $f$. Specifically, given a dataset of $N$ tasks $\mathcal{D} = \{(\phi_i, G_i)\}_{i=1}^N$, we address the following two tasks:

- learning an algorithm $\mathcal{A}_\theta : (\phi, G) \mapsto f$ parameterized by $\theta$ that can find the function $f_i$ for $(\phi_i, G_i) \in \mathcal{D}$;
- given a new task set $\mathcal{D}'$, adapt the above learned algorithm $\mathcal{A}_\theta$ and execute it on new tasks in $\mathcal{D}'$.

This setting poses two difficulties in learning. First, the ground truth target function $f$ is not readily available, making it difficult to formulate as a supervised learning problem. Second, the constraint

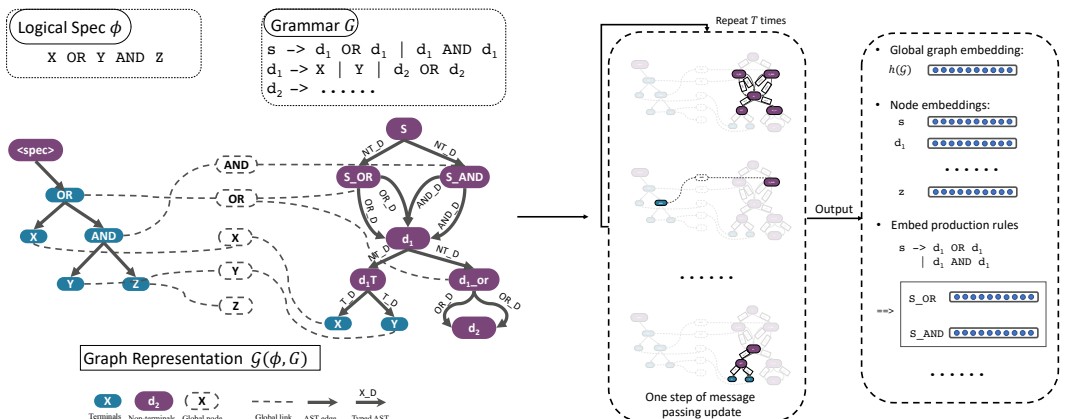

Figure 1: Graph representation of syntax and semantic constraint. Note that the reversed links are also added in our representation.

$\phi$ is typically verified using an SAT or SMT solver, and this solver in turns expects the generated $f$ to be complete. This means the weak supervision signal will only be given after the entire program is generated. Thus, it is natural to formulate $\mathcal{A}_\theta$ as a reinforcement learning algorithm. Since each instance $(\phi_i, G_i) \in \mathcal{D}$ is an independent task with different syntactic and semantic constraints, the key to success is the design of such meta-learner, which we elaborate in Sec 3.

## 3 META-SOLVER MODEL

This section presents our meta-solver model for solving the two problems formulated in Sec 2. We first introduce formal notation in Sec 3.1. To enable the transfer of knowledge across tasks with different syntactic and semantic constraints, we propose a representation framework in Sec 3.2 to jointly encode the two kinds of constraints. The representation needs to be general enough to encode constraints with different specifications. Lastly, we introduce the Grammar Adaptive Policy Network in Sec 3.3 that executes a program generation policy while automatically adapting to different grammars encoded in each task specification.

### 3.1 FORMAL DEFINITIONS

We formally define key concepts in the SyGuS problem formulation as follows.

**semantic spec** $\phi$: The spec itself is a program written using some grammar. In our case, the grammar used in spec $\phi$ is different from the grammar $G$ that specifies the syntax of the output program. However, in many practical cases the tokens (i.e., the dictionary of terminal symbols) may be shared across the input spec and the output program.

**CFG** $G$: A context free grammar (CFG) is defined as $G = \langle \mathcal{V}, \Sigma, \mathcal{R}, s \rangle$. Here $\mathcal{V}$ denotes the non-terminal tokens, while $\Sigma$ represents the terminal tokens. $s$ is a special token that denotes the start of the language, and the language is generated according to the production rules defined in $\mathcal{R}$. For a given non-terminal, the associated production rules can be written as $\alpha \rightarrow \beta_1 | \beta_2 \ldots | \beta_{n_\alpha}$, where $n_\alpha$ is the branching factor for non-terminal $\alpha \in \mathcal{V}$, and $\beta_i = u_1 u_2 \ldots u_{|\beta_i|} \in (\mathcal{V} \bigcup \Sigma)^*$. Each production rule $\alpha \rightarrow \beta_i \in \mathcal{R}$ represents a way of expanding the grammar tree, by attaching nodes $u_1, u_2, \ldots, u_{|\beta_i|}$ to node $\alpha$. The expansion is repeated until all the leaf nodes are terminals.

**Output function** $f$: The output is a program in the language generated by $G$. A valid output $f$ must satisfy both the syntactic constraints specified by $G$ and the semantic constraints specified by $\phi$.

### 3.2 TASK REPRESENTATION

Different from traditional neural program synthesis tasks, where the program grammar and vocabulary is fixed, each individual task in our setting has its own form of grammar and semantic specification. Thus in the program generation phase (which we will elucidate in Sec 3.3), one cannot assume a fixed CFG and use a tree decoder like in Kusner et al. (2017) and Bunel et al. (2018). To enable

such generalization across different grammars, the information about the CFG for each task needs to be captured in the task representation.

Since the semantic spec program $\phi$ and the CFG $G$ have rich structural information, it is natural to use graphs for their representation. Representing the programs using graphs has been successfully used in many programming language domains. In our work, we further extend the approach by Allamanis et al. (2018b) with respect to the following aspects:

- Instead of only representing the semantic spec program $\phi$ as a graph, we propose to jointly represent it along with the grammar $G$.

- To allow information exchange between the two graphs, we leverage the idea of Static Single Assignment (SSA) form in compiler design. That is, the same variable (token) that may be assigned (defined) at many different places should be viewed differently, but on the other hand, these variations correspond to the same original thing. Specifically, we introduce global nodes for shared tokens and global links connecting these globally shared nodes and local nodes that (re)define corresponding tokens.

The overall representation framework is described in Figure 1. To construct the graph, we first build the abstract syntax tree (AST) for the semantic spec program $\phi$, according to its own grammar (typically different from the output grammar $G$). To represent the grammar $G$, we associate each symbol in $\mathcal{V} \bigcup \Sigma$ with a node representation. Furthermore, for a non-terminal $\alpha$ and its corresponding production rules $\alpha \to \beta_1 | \beta_2 \dots | \beta_{n_\alpha}$, we create additional nodes $\alpha_i$ for each substitute $\beta_i$. The purpose is to enable grammar adaptive program generation, which we elaborate in Sec 3.3. As a simplification, we merge all nodes $\alpha_i$ representing $\beta_i$ that is a single terminal token into one node. Finally, the global nodes for shared tokens in $\Sigma$ are created to link together the shared variable and operator nodes. This enables information exchange between the syntactic and semantics specifications.

To encode the joint graph $\mathcal{G}(\phi, G)$, we use graph neural networks to get the vector representation for each node in the graph. Specifically, for each node $v \in \mathcal{G}$, we use the following parameterization for one step of message passing style update:

$$h_v^{t+1} = \text{Aggregate}\Big( \{ F(h_u^t, e_{u,v}) \}_{u \in \mathcal{N}(v)} \Big) \tag{1}$$

Lastly, $\{h_v^T\}_{v \in \mathcal{G}}, h_v^T \in \mathbb{R}^d$ are the set of node embeddings. Here $\mathcal{N}(v)$ is the set of neighbor nodes of $v$, and $e_{u,v}$ denotes the type of edge that links the node $u$ and $v$. We parameterize $F$ in a way similar to GGNN (Li et al., 2015), i.e., $F(h^t, e) = \sigma(W_t^{e\top} h^t)$ where we use different matrices $W \in \mathbb{R}^{d \times d}$ for different edge types and different propagation steps $t$. We sum over all the node embeddings to get the global graph embedding $h(\mathcal{G})$.

In addition to the node embeddings and global graph embedding, we also obtain the embedding matrix for each non-terminal node. Specifically, given node $\alpha$, we will have the embedding matrix $H_\alpha \in \mathbb{R}^{n_\alpha \times d}$, where the $i$th row of $H_\alpha^{(i)}$ is the embedding of node $\alpha_i$ that corresponds to substitution $\beta_i$. This enables the grammar adaptive tree expansion in Sec 3.3.

### 3.3 GRAMMAR ADAPTIVE POLICY NETWORK

To enable the meta-solver to generalize across different tasks, both the task representation and program generation policy should be shared. We perform task conditional program generation for this purpose. Overall the generation is implemented using tree recursive generation, in the depth-first search (DFS) order. However, to handle different grammars specified in each task, we propose to use the grammar adaptive policy network. The key idea is to make the policy parameterized by decision embedding, rather than a fixed set of parameters. This mechanism is inspired by the pointer network (Vinyals et al., 2015) and graph algorithm learning (Dai et al., 2017).

Specifically, suppose we are at the decision step $t$ and try to expand the non-terminal node $\alpha_t$. For different tasks, the non-terminals may not be the same; furthermore, the number of ways to expand a certain non-terminal can also be different. As a result, we cannot simply have a parameterized layer $W^\top h_{\alpha_t}$ to calculate the logits of multinomial distribution. Rather, we use the embedding matrix $H_{\alpha_t} \in \mathbb{R}^{n_{\alpha_t} \times d}$ to perform decision for this time step. This embedding matrix is obtained as described in Sec 3.2.

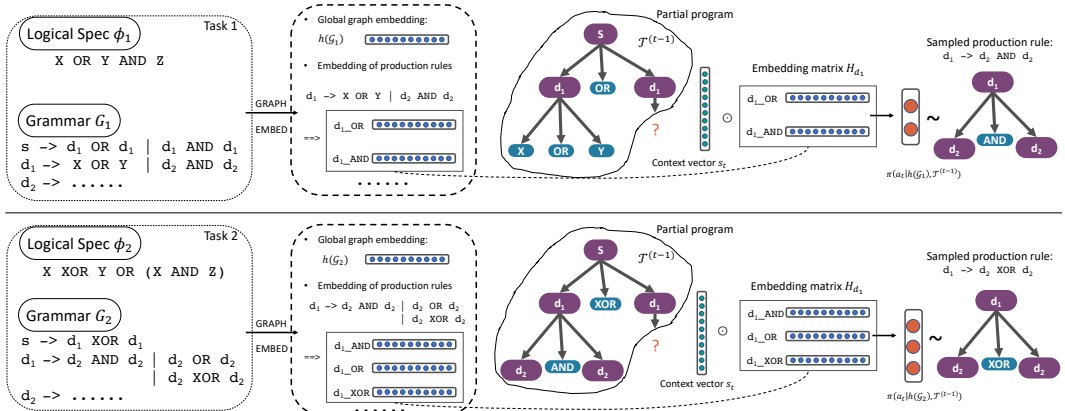

Figure 2: Generating solution using the grammar adaptive policy network. This figure shows one step of policy roll-out, which demonstrates how the same policy network handles different tasks with different grammar $G_1$ and $G_2$.

Now we are able to build our policy network in an auto-regressive way. Specifically, the policy $\pi(f|\phi, G)$ can be parameterized as:

$$\pi(f|\phi, G) = \prod_{t=1}^{|f|} \pi(a_t|h(\mathcal{G}), \mathcal{T}^{(t-1)}), \text{ where } \mathcal{T}^{(t-1)} = \alpha_1 \dots \alpha_{t-1} \text{ denotes the partial tree} \quad (2)$$

Here the probability of each action (in other words, each tree expansion decision) is defined as $\pi(a_t|h(\mathcal{G}), \mathcal{T}^{(t-1)}) \propto \exp(H_\alpha^{(i)\top} s_t)$, where $s_t \in \mathbb{R}^d$ is the context vector that captures the state of $h(\mathcal{G})$ and $\mathcal{T}^{(t-1)}$. In our implementation, $s_t$ is tracked by a LSTM decoder whose hidden state is updated by the embedding of the chosen action $h_{\alpha_t}$. The initial state $s_0$ is obtained by passing graph embedding $h(\mathcal{G})$ through a dense layer with matching size.

## 4 SOLVING VIA REINFORCEMENT LEARNING

In this section, we present a reinforcement learning framework for the meta-solver. Formally, let $\theta$ denote the parameters of graph embedding and adaptive policy network. For a given pair of instances $(\phi, G)$, we learn a policy $\pi_\theta(f|\phi, G)$ parameterized by $\theta$ that generates $f$ such that $\phi \equiv f$.

**Reward design**: The RL episode starts by accepting the representation of tuple $\langle \phi, G \rangle$ as initial observation. During the episode, the model executes a sequence of actions to expand non-terminals in $f$, and finishes the episode when $f$ is complete. Upon finishing, the SAT solver is invoked and will return a binary flag indicating whether $f$ satisfies $\phi$ or not. An obvious reward design would be directly using the binary value as the episode return. However, this leads to a high variance in returns as well as a highly non-smooth loss surface. Here, we propose to smooth the reward as follows: for each specification $\phi$ we maintain a test case buffer $B_\phi$ that stores all input examples observed so far. Each time the SAT solver is invoked for $\phi$, if $f$ passes then a full reward of 1 is given, otherwise the solver will generate a counter-example $b$ besides the binary flag. We then sample interpolated examples around $b$ which we denote the set as $\hat{B}_b$. Then the reward is given as the fractions of examples in $B_\phi$ and $\hat{B}_b$ where $f$ has the equivalent output as $\phi$

$$r = \frac{\sum_{b \in B_\phi \cup \hat{B}_b}[f(b) \equiv \phi(b)]}{|B_\phi \cup \hat{B}_b|}.$$

At the end of the episode, the buffer is updated as $B_\phi \leftarrow B_\phi \cup \hat{B}_b$ for next time usage. In the extreme case where all inputs can be enumerated, e.g. binary or discrete values, it reduces to computing the fraction of passed examples over the entire test case set. This is implemented in our experiment on the cryptographic circuit synthesis task.

### 4.1 LEARNING THE META-SOLVER

In the meta-learning setting, the framework learns to represent a set of different programs and navigate the generation process under different constraints. We utilize the Advantage Actor-Critic (A2C) for model training. Given a training set $\mathcal{D}$, a minibatch of instances are sampled from $\mathcal{D}$ for each epoch. For each instance $\langle \phi_i, G_i \rangle$, the model performs a complete rollout using policy $\pi_\theta(f|\phi_i, G_i)$. The actor-critic method computes the gradients w.r.t to $\theta$ of each instance as

$$d\theta \leftarrow \sum_{t=1}^{|f|} \nabla_\theta log\pi(\alpha_t|h(\mathcal{G}), \mathcal{T}^{(t)})(\gamma^{|f|-t}r - V(s_t; \omega)),$$

where $\gamma$ denotes the discounting factor and $V(s_t; \omega)$ is a state value estimator parameterized by $\omega$. In our implementation, this is modeled as a standard MLP with scalar output. It is learned to fit the expected return, i.e., $\min_\omega \mathbb{E} \left\| \sum_{t=1}^{|f|} \gamma^t r - V(s_t; \omega) \right\|$. Gradients obtained from each instance are averaged over the minibatch before applying to the parameter.

## 5 EXPERIMENTS

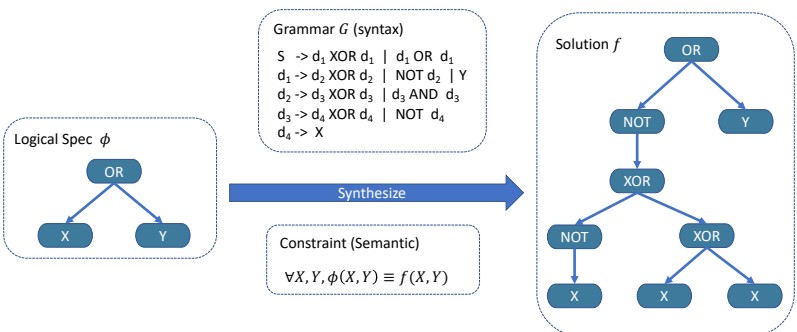

Figure 3: An example of a circuit synthesis task from the 2017 SyGuS competition. Given the original program specification which is represented as an abstract syntax tree (left), the solver is tasked to synthesize a new circuit $f$ (right). The synthesis process is specified by the syntactic constraint $G$ (top), and the semantic constraint (bottom) specifies that $f$ must have functionality equivalent to the original program.

We evaluate the our framework[1] on cryptographic circuit synthesis tasks (Eldib et al., 2016) which constitute a challenging benchmark suite from the general track of the SyGuS Competition (2017). The dataset contains 214 tasks, each of which is a pair of logical specification, describing the correct functionality, and a context free grammar, describing the timing constraints for input signals. The goal is to find an equivalent logical expression which is required to follow the given context free grammar in order to avoid potential timing channel vulnerabilities. Figure 3 shows an illustrative example. Each synthesis task has a different logical specification as well as timing constraints, and both the logical specification and context free grammar varies from task to task, posing a significant challenge in representation learning. As a result, this suite of tasks serves as an ideal testbed for our learning framework and its capability to generalize to unseen specifications and grammars.

The experiments are conducted in two learning settings. First, we test our framework as an out-of-box solver, which means the training set $\mathcal{D}$ and testing set $\mathcal{D}'$ are the same and contain only one instance. In other words, the framework is tasked to solve only one instance at a time. This test-on-train setting serves to investigate the capacity of our framework in representation and policy learning, as the model can arbitrarily "exploit" the problem without worrying about overfitting. This setting also enables us to compare our framework to classical solvers developed in the formal methods community. As those solvers do not utilize learning-based strategies, it is sensible to also limit our framework not to carry over prior knowledge from a separate training set.

Second, we evaluate the model as a meta-solver which is trained over a training set $\mathcal{D}$, and finetuned on each of the new tasks in a separate set $\mathcal{D}'$. In this setting, we aim to demonstrate that our

---

[1]Our code and data are available on GitHub: `https://github.com/PL-ML/metal`

Table 1: Number of instances solved using: 1) EUSolver, 2) CVC4, 3) ESymbolic, and 4) Out-of-Box Solver. For each solver, the maximum time in solving an instance and the average and median time over all solved instances are also shown below.

| | # instances solved | Max time | Avg time | Median time |
|---|---|---|---|---|
| EUSolver | 153 / 214 | 1h39m | 3m | 3s |
| CVC4 | 129 / 214 | 5h50m | 30m | 6s |
| ESymbolic | 31 / 214 | 40m | 8m | 5m |
| Out-of-Box Solver | 141 / 214 | 4h11m | 33m | 3m |

framework is capable of learning a transferable representation and policy in order to efficiently adapt to unseen tasks.

## 5.1 LEARNING AN OUT-OF-BOX SOLVER

In the out-of-box solver setting, we compare our framework against solvers built based on two classical approaches: a SAT/SMT constraint solving based approach and a search based approach. For the former, we choose CVC4 (Reynolds et al., 2015), which is the state-of-the-art SMT constraint solver; for the latter, we choose EUSolver (Alur et al., 2017b), which is the winner of the SyGuS 2017 Competition (Alur et al., 2017a). Furthermore, we build a search based solver as baseline, ESymbolic, which systematically expands non-terminals in a predefined order (e.g. depth-first-search) and effectively prunes away partially generated candidates by reducing it to 2QBF (Balabanov et al., 2016) satisfiability check. ESymbolic can be viewed as a generalization of EUSolver by replacing the carefully designed domain-specific heuristics (e.g. indistinguishability and unification) with 2QBF.

In order to make the comparison fair, we run all solvers on the same platform with a single core CPU available[2], even though our framework could take advantage of hardware accelerations, for instance, via GPUs and TPUs. We measure the performance of each solver by counting the number of instances it can solve given a 6 hours limit spent on each task. It is worth noting that comparing running time only gives a limited view of the solvers' performance. Although the hardware is the same, the software implementation can make many differences. For instance, CVC4 is carefully re-designed and re-implemented in C++ as the successor of CVC3 (Barrett et al., 2003), which has been actively improved for more than a decade. To our best knowledge, the design and implementation of EUSolver is directly guided by and heavily tuned according to SyGuS benchmarks. In contrast, our framework is a proof-of-concept prototype implemented in Python and has not yet been tuned for running time performance.

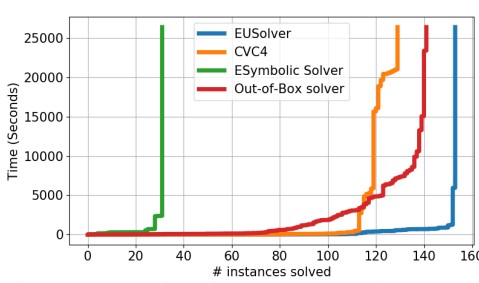

Figure 4: Running time cost by each solver.

In Table 1, we summarize the total number of instances solved by each solver as well as the maximum, average and median running time spent on solved instances. In terms of the absolute number of solved instances, our framework is not yet as good as EUSolver, which is equipped with specialized heuristics. However, EUSolver fails to solve 4 instances that are only solved by our framework. All instances solved by CVC4 and ESymbolic are a strict subset of instances solved by EU-Solver. Thus, besides being a new promising approach, our framework already plays a supplementary role for improving the current state-of-the-art. Compared with the state-of-the-art CVC4 solver, our framework has smaller maximum time but higher average and median time usage. This suggests that our framework excels at solving difficult instances with better efficiency. This observation is further confirmed in Figure 4, where we plot the time usage along with the number of instances solved. This suggests that canonical solvers such as CVC4 are efficient in solving simple instances, but have inferior scalability compared to our dynamically adapted approach when

---

[2]The SyGuS 2017 competition gives each solver 4-core 2.4GHz Intel processors with 128 GB memory and wallclock time limit of 1 hour; our evaluation uses AMD Opteron 6220 processor, assigns each solver a single core with 32 GB memory and wallclock time limit of 6 hours.

| # candidates | Percentage solved | | |
|---|---|---|---|
| generated | 20% | 40% | 60% |
| Out-of-Box Solver | 2564 | 18K | 102K |
| Meta-Solver | 205 | 4.5K | 59K |
| Reduction | 12.5× | 4.0× | 1.7× |

(a)

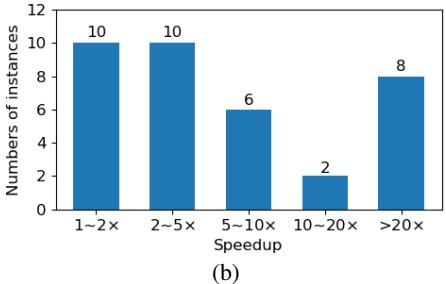

(b)

Figure 5: Performance improvement with meta-learning. (a) Accumulated number of candidates generated in order to solve 20%, 40%, and 60% of the testing tasks; and (b) speedup distribution over individual instances.

the problem becomes more difficult, where we can see a steeper increase in time usages by CVC4 in solving 110 and more instances. Though EUSolver has superior scalability, it is achieved by a number of heuristics that are manually designed and iteratively improved by experts with the same benchmark on hand. In contrast, our framework learns a policy to solve hard instances from scratch on the fly without requiring training data at all.

### 5.2 LEARNING ACROSS DIFFERENT TASKS

We next evaluate whether our framework is capable of learning transferable knowledge across different synthesis tasks. We randomly split the 214 circuits synthesis tasks into two sets: 150 tasks for training and the rest 64 tasks for testing. The meta-solver is then trained on the training set for 35000 epochs using methods introduced in Sec 4.1. For each epoch, a batch of 10 tasks are sampled. The gradients of each task are averaged and applied to the model parameters using Adam optimizer. In testing phase, the trained meta-solver is finetuned on each task in the testing set until either a correct program is synthesized or timeout occurs. This process is similar to the setting in Sec 5.1 but with smaller learning rate and exploration.

We compare the trained meta-solver with the out-of-box solver in solving tasks in the test set. Out of 64 testing tasks, the out-of-box solver and meta-solver can solve 36 and 37 tasks, respectively. Besides the additional task solved, the performance is also greatly improved by meta-solver, which is shown in Figure 5. Table 5(a) shows the accumulated number of candidates generated to successfully solve various ratios of testing tasks. We see that the number of explored candidates by meta-solver is significantly reduced: for 40% of testing tasks (i.e., 66% of solved tasks), meta-learning enable 4x reduction on average. The accumulated reduction for all solved tasks (60% of testing tasks) is not that significant. This is because meta-learning improve dramatically for most (relatively) easy tasks but helps slightly for a few hard tasks, which actually dominate the number of generated candidates. Figure 5(b) shows the speedup distribution over the 36 commonly solved tasks. Meta-solver achieves at least 2x speedup for most benchmarks, orders of magnitude improvement for 10 out of 36 unseen tasks, and solves one task that is not solvable without meta-learning.

## 6 RELATED WORK

We survey work on symbolic program synthesis, neural program synthesis, and neural induction.

**Symbolic program synthesis.** Automatically synthesizing a program from its specification was first posed by Manna & Waldinger (1971). It received renewed attention with advances in SAT and SMT solvers (Solar-Lezama et al., 2006) and found application in problems in various domains as surveyed by Gulwani et al. (2017). In this context, SyGuS (Alur et al., 2013) was proposed as a common format to express these problems. Several implementations of SyGuS solvers exist, including by constraint solving (Reynolds et al., 2015), divide-and-conquer (Alur et al., 2017b), and stochastic MCMC search (Schkufza et al., 2013), in addition to various domain-specific algorithms. A number of probabilistic techniques have been proposed to accelerate these solvers by modeling syntactic aspects of programs. These include PHOG (Lee et al., 2018), log-bilinear tree-traversal models (Maddison & Tarlow, 2014), and graph-based statistical models (Nguyen & Nguyen, 2015).

**Neural program synthesis.** Several recent works have used neural networks to accelerate the discovery of desired programs. These include DeepCoder (Balog et al., 2017), Bayou (Murali

et al., 2018), RobustFill (Devlin et al., 2017), Differentiable FORTH (Bošnjak et al., 2017), neuro-symbolic program synthesis (Parisotto et al., 2016; Bunel et al., 2018), neural-guided deductive search (Vijayakumar et al., 2018), learning context-free parsers (Chen et al., 2018), and learning program invariants (Si et al., 2018). The syntactic specification in these approaches is fixed by defining a domain-specific language upfront. Also, with the exception of Si et al. (2018), the semantic specification takes the form of input-output examples. Broadly, these works have difficulty with symbolic constraints, and are primarily concerned with avoiding overfitting, coping with few examples, and tolerating noisy examples. Our work relaxes both these kinds of specifications to target the general SyGuS formulation. Recently Ellis et al. (2018) propose gradually bootstrapping domain-specific languages for neurally-guided Bayesian program learning, while our work concerns learning programs that use similar grammars, which may or may not be incremental.

**Neural program induction.** Another body of work includes techniques in which the neural network is itself the computational substrate. These include neural Turing machines (Graves et al., 2014) that can learn simple copying/sorting programs, the neural RAM model (Kurach et al., 2016) to learn pointer manipulation and dereferencing, the neural GPU model (Kaiser & Sutskever, 2015) to learn complex operations like binary multiplication, and Cai et al. (2017)'s work to incorporate recursion. These approaches have fundamental problems regarding verifying and interpreting the output of neural networks. In contrast, we propose tightly integrating a neural learner with a symbolic verifier so that we obtain the scalability and flexibility of neural learning and the correctness guarantees of symbolic verifiers.

# 7 CONCLUSION

We proposed a framework to learn a transferable representation and strategy in solving a general formulation of program synthesis, i.e. syntax-guided synthesis (SyGuS). Compared to previous work on neural synthesis, our framework is capable of handling tasks where 1) the grammar and semantic specification varies from task to task, and 2) the supervision is weak. Specifically, we introduced a graph neural network that can learn a joint representation over different pairs of syntactic and semantic specifications; we implemented a grammar adaptive network that enables program generation to be conditioned on the specific task; and finally, we proposed a meta-learning method based on the Advantage Actor-Critic (A2C) framework. We compared our framework empirically against one baseline following a similar search fashion and two classical synthesis engines. Under the out-of-box solver setting with limited computational resources and without any prior knowledge from training, our framework is able to solve 141 of 214 tasks, significantly outperforming the baseline ESymbolic by 110. In terms of the absolute number of solved tasks, the performance is comparable to two state-of-the-art solvers, CVC4 and EUSolver, which solve 129 and 153 respectively. However, the two state-of-the-art solvers failed on 4 tasks solved by our framework. When trained as a meta-solver, our framework is capable of accelerating the solving process by $2\times$ to $100\times$.

## ACKNOWLEDGMENTS

We thank the anonymous reviewers for their insightful comments. This research was supported in part by NSF awards #1836936 and #1836822.

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
