# OpenReview forum: "Learning a Meta-Solver for Syntax-Guided Program Synthesis"
_ICLR.cc/2019/Conference_

### Official Review · AnonReviewer1 · 2018-11-01
**Generating (syntactic and functional) specification-satisfying programs via Reinforcement Learning**

**Rating:** 7
**Confidence:** 2

**Review:**

The authors design a program synthesizer that tries to satisfy per-instance specific syntactic and functional constraints,
based on sampling trajectories from an RL agent that at each time-step expands a partial-program.

The agent is trained with policy gradients with a reward shaped as the ratio of input/output examples that the synthesized program satisfies.

With the 'out-of-box' evaluation, the authors show that their agent can explore more efficiently the harder problems than their non-learning alternatives even from scratch.
(My intuition is that the agent learns to generate the most promising programs)
It would be good to have a Monte Carlo Tree Search baseline on the'out-of-box' evaluation, to detect exploration exploitation trade-offs.

The authors show with the 'meta-solver' approach that the agent can generalize to and also speed up unseen (albeit easy-ish in the authors words) instances.

Clarity: Paper is clear and nicely written.

Significance: Imagine a single program synthesizer that could generate C++/Java/Python/DSLs  programs and learn from all its successes and failures! This is a step towards that.

Pros:
+ Generating spec-following programs for different grammars.
+ partial tree expansion takes care of syntactic constraints.
Neutral
· The grammar and specification diversity may be too low to feel impressive.
· It would have been nicer by computing likelihood for unseen instances with unique and known solutions (that is, without finetuning).
Cons:
- No Tree Search baseline.
- No results on programs with control flow/internal state.

---

> ### Author Response · Authors · 2018-11-19
> **Response to Reviewer 1**
>
> We appreciate your insightful review comments. We address the concerns and questions as follows:
>
> >> Have you considered any tree search baseline, for example, Monte-Carlo Tree Search?
>
> In our evaluation, the ESymbolic baseline is a tree search method, except that it expands the nonterminals in a deterministic depth-first fashion and does pruning using constraint solving (e.g. 2QBF) along the way. For the proposed method, however, while the generated program that our model operates on indeed can be represented by a tree, the RL algorithm we use is essentially model-free, i.e. it is agnostic to the transition dynamics. We agree with the reviewer that this approach can be further improved with a model-based approach such as MCTS, since we can track the dynamics easily, and presumably yields better performance than the current purely model-free approach. On the other hand, as one of the main motivations of our work is to study how to cast the classical problem into a learning task, we have been focused on the comparison between learning and non-learning methods, instead of model-free and model-based methods. However, it would be definitely interesting to explore more on the model-based methods for program synthesis, and we leave this to our future work.
>
> >>  How about generalization without fine-tuning?
>
> Indeed, it would be great to generalize to unseen programs even without fine-tuning, but in the meta-learning setting, it is typically very hard as it requires a lot of samples not only in the data space but also in the task space, for which we only have around 200 tasks. We did test the performance of the learner without fine-tuning, and, with no surprise, it turns out to perform worse than the out-of-the-box version.
>
> On the other hand, this train-and-finetune fashion is becoming widely accepted by a number of recent works on meta-reinforcement-learning, for instance, “Recasting Gradient-Based Meta-Learning as Hierarchical Bayes”.
>
> >> Programs seems too low level and lacks of control flow/internal state, which are common features in general programming language like C, Java, Python, etc.
>
> This is a great suggestion for our future work. We believe learning programs from logical specifications in a general programming language is an important direction in artificial intelligence, and our work is a step towards this direction.

---

### Official Review · AnonReviewer3 · 2018-11-03
**Interesting technique for a challenging synthesis domain, but some details are not clear**

**Rating:** 7
**Confidence:** 5

**Review:**

This paper presents a reinforcement learning based approach to learn a search strategy to search for programs in the generic syntax-guided synthesis (SyGuS) formulation. Unlike previous neural program synthesis approaches, where the DSL grammar is fixed or the specification is in the form of input-output examples only, the SyGuS formulation considers different grammars for different synthesis problems and the specification format is also more general. The main idea of the approach is to first learn a joint representation of the specification and grammar using a graph neural network model, and then train a policy using reinforcement learning to guide the search with a grammar adaptive policy network that is conditioned on the joint representation. Since the specifications considered here are richer logical expressions, it uses a SAT solver for checking the validity of the proposed solution and to also obtain counterexamples for future rewards. The technique is evaluated on 210 SyGuS benchmarks coming from the cryptographic circuit synthesis domain, and shows significant improvements in terms of number of instances solved compared to CVC4 and ESymbolic baseline search techniques from the formal methods community. Moreover, the learnt policy is also showed to generalize beyond the benchmarks on which it is trained and the meta-solver performs reasonably well compared to the per-task out-of-box solver.

Overall, this paper tackles a more challenging synthesis problem than the ones typically considered in recent neural synthesis approaches. The previous synthesis approaches have mostly focused on learning programs in a fixed grammar (DSL) and with specifications that are typically based on either input-output examples or natural language descriptions. In the SyGuS formulation, each task has a different grammar and moreover, the specifications are much richer as they can be arbitrary logical expressions on program variables. The overall approach of using graph neural networks to learn a joint representation of grammars with the corresponding logical specifications, and then using reinforcement learning to learn a search policy over the grammar is quite interesting and novel. The empirical results on the cryptographic benchmarks compare favorably to state of the art CVC4 synthesis solver.

However, there were some details in the model description and evaluation that were not very clear in the current presentation.

First, the paper mentions that it uses the idea of Static Single Assignment (SSA) form for the graph representation. What is the SSA form of a grammar and of a specification?

It was also not very clear how the graphs are constructed from the grammar. For example, for the rule d1 -> X OR Y | d2 OR d2 in Figure 1, are there two d_OR nodes or a single node d_OR shared by both the rules? Similarly, what is the d_T node in the figure? It would be good to have a formal description of the nodes and edges in the graph constructed from the spec and grammar.

Since the embedding matrix H_d can be of variable size (different sizes of expansion rules), it wasn’t clear how the policy learns a conditional distribution over the variable number of actions. Is there some form of padding of the matrix and then masking being used?

For the reward design, the choice of using additional examples in the set B_\phi was quite interesting. But there was no discussion about how the interpolation technique works to generate more examples around a counterexample. Can you provide some more details on how the interpolation is being performed?

Also, how many examples were typically used in the experiments? It might be interesting to explore whether different number of examples lead to different results. How does the learning perform in the absence of these examples with the simple binary 0/1 reward?

From last year’s SyGuS competition, it seems that the EUSolver solves 152 problems from the set of 214 benchmarks (Table 4 in http://sygus.seas.upenn.edu/files/SyGuSComp2017.pdf). For the evaluation, is ESymbolic baseline solver different that the EUSolver? Would it be possible to evaluate the EUSolver on the same hardware and timeout to see how well it performs on the 210 benchmarks?

The current transfer results are only limited to the cryptographic benchmarks. Since SyGuS also has benchmarks in many other domains, would it be interesting to evaluate the policy transfer to some other non-cryptographic benchmark domain?

---

> ### Author Response · Authors · 2018-11-19
> **Response to Reviewer 3**
>
> We appreciate your effort in providing detailed and helpful reviews. We address the concerns and questions as follows:
>
> >> Could you clarify the SSA form for graph representation?
> The graph representation is roughly inspired by the so-called static single assignment: though the same variable is assigned and used at many places, they can be distinguished by attaching a subscript at each place it is assigned. We view the same logical operator used in different grammar rules as slightly different ones, but they do have the same semantic meaning. So we create separate nodes for the same logical operator in different grammar rules, but also introduce a corresponding global node, which is intended to summarize its effects in different rules.  Given that SSA is simply an analogy rather than a formal notion for grammar and specification, we would like to give more intuitive names (e.g. global node and global link) for the current SSA node and SSA link in the graph representation.
>
> >> Typos in figure-1.
> Thanks for pointing this out, and we apologize for the typos in figure-1.  The rule d1 -> X OR Y | d2 OR d2  is meant to be d1 -> X | Y | d2 OR d2. In the case where two OR derivations are indeed given, there would be two d_OR nodes.  And, d_T is used to indicate that X and Y are terminals.
>
> >> How the policy learns a conditional distribution over the variable number of actions. Is there some form of padding of the matrix and then masking being used?
> When choosing the action, we perform dot product between the state vector and each row of the H_{\alpha_t}, which yields a n_{\alpha_t}-dimensional vector, where n_{\alpha_t} is the number of possible expansions. Then we take the softmax over this vector, which gives the multinomial over actions. This is similar to an attention mechanism. Therefore, no additional parameter or padding is needed to handle the variable number of actions.
>
> >> How is the interpolation being performed? Also, how many examples were typically used in the experiments? It might be interesting to explore whether different number of examples lead to different results. How does the learning perform in the absence of these examples with the simple binary 0/1 reward?
> Interpolation is more straightforward in the domain where numerical values are involved. For the domain in our evaluation, which contains only Boolean values, by interpolation we mean randomly flipping the truth value of some variable of an example to get a new example. We view interpolation as an approximation to the exhaustive enumeration; reward obtained with more interpolated samples will certainly be more reliable than that obtained with less samples. One extreme case is to keep a single sample at a time, which is essentially the simple binary 0/1 reward. We ran the experiment as the reviewer suggested, and out-of-box solver with 0/1 reward can solve 122 tasks.  In terms of the number of examples, typically, 200 (or less) examples are used for each task.
>
> >> Can you please run EUSolver using your setup?
> As suggested by the reviewer, we have run EUSolver with the same setup used in our evaluation. It solves 153 tasks (1 more task is solved in contrast with the SyGus 2017 report). These solved tasks are strictly a superset of tasks solved by CVC4 and ESymbolic. But EUSolver fails to solve 4 tasks solved by our framework. In terms of absolute number of solved tasks, our framework is not yet as good as EUSolver, but it provides a new and complementary way to SyGus tasks. We have incorporated this discussion in our revision.
>
> In terms of comparison with the state-of-the-art, we favored CVC4 solver rather than EUSolver, because CVC4 is a general SMT solver, while EUSolver is designed as a collection of specialized heuristics (e.g. indistinguishability and unification) for each benchmark category of SyGus competition, and (to our best knowledge) its design and implementation are guided and heavily tuned according to SyGus benchmarks. Our framework is also a general solver without requirement for specialized heuristics for each domain. The speciality of EUSolver motivates us to develop a more general solver as baseline, namely ESymbolic, by replacing domain-specific heuristics used in EUSolver with a more general heuristic (i.e. partial program pruning with QBF).
>
> >> How about other categories in SyGus competition?
> The other categories are not included in our evaluation due to two reasons. First, they have a very few number of tasks, most of which is around 30 or even less. Second, most tasks only have a few input/output example pairs, rather than a logical formal specification that is necessary for our approach to draw counterexamples.

---

> ### Comment · Area_Chair1 · 2018-11-30
> **Please consider author response**
>
> Reviewer 3, the authors of this paper have submitted a fairly detailed response to your own detailed review (thanks for that!). It is important that there be some consideration of their reply, and if needed, discussion. Please take the time to review and respond to their rebuttal, and either reconsider your assessment or explain why you stand by it in its current form.

---

### Official Review · AnonReviewer2 · 2018-11-03
**Good paper**

**Rating:** 7
**Confidence:** 4

**Review:**

This paper presents a (meta-)solver for particular program synthesis problems, where the model has access to a (logic) specification of the program to be synthesized, and a grammar that can change from one task instance to another. The presented model is an RL-based model that jointly trains 1) the joint graph-based embedding of the specification and the grammar, and 2) a policy able to operate on different (from instance to instance) grammars. Interestingly, not only can the model operate as a stand-alone solver, but it can be run as a meta-solver - trained on a subset of tasks, and applied (with tuning) on a new task. Experiments show that the model outperforms two baselines (one being a (near-to-)SOTA model) in the stand-alone setting and that the model successfully transfers knowledge (considers fewer candidates) in the meta-solving mode.

First, I enjoyed reading the paper. I think the problem is interesting, particularly due to the model being able to train and operate on various grammars (from task to task), and not on a single, pre-specified grammar. The additional bonus is that the problem the paper solves does not require program as supervision, but an external verifier.
The evaluation shows that this approach not only makes sense but (significantly) outperforms, under same conditions, specialized program synthesis programs. However, there’s one issue here, and that’s what the comparison hasn’t been done to SOTA model but to a less performant model (see issues).
The particular approach of jointly training a specification+grammar graph embedding and learning a policy that acts on different grammars seems original and significant enough for publication.
The paper is well (with a few kinks) written, and mostly clear. There are still some issues in the paper.

Issues:
- The dataset used is 210 cryptographic circuit synthesis tasks from SyGuS 2017. Why only this particular subset of all the tasks, and not the other tasks/categories (there is 569 of them in total, no)?
- Alur et al mention 214 examples in the said tasks, yet the paper says 210. Why?
- The SyGuS results paper https://arxiv.org/abs/1711.11438 mentions EUSolver as the SOTA model, solving 152 tasks (out of 214). Why didn’t you compare your model to EUSolver?
- The same paper reports CVC4 solving 117 tasks (out of 214), as opposed to 129 (out of 210) reported in your paper. Could you comment on the (possible) differences in the experimentation protocol?
- you mention global graph embedding, but you never describe how you calculate it
- abstract mentions outperforming two SOTA engines, but later you say ESymbolic is a baseline (which it seems by description)

Questions:
- W for different edge types and different propagation steps t? Why is there a need for such a large number of parameters? What is the number of propagation steps?
- In the extreme case where all inputs can be enumerated - how often does this happen in the tasks you solve?
- figure 2 is not clear. There is too much information on one side (grammar) and too little on the other (what is the meaning of \tau^(t-1)?)? Is the tree on the right a generated subtree?
- details of the state s are unclear - it is tracked by an LSTM? Is there a concrete training signal for s, or is it a part of the architecture and everything is end-to-end trainable from the final reward? The same for s0=MLP(h(G)) - is that also trained in the same way?
- can you provide some intuition on why you chose that particular architecture (state-tracking LSTM,  s0 as such, instead of something simpler?)
- can you provide details on the state value estimator MLP architecture, as well as the s0 MLP, and the state-tracking LSTM?
- the probability of each action (..) is defined as ….H_\alpha^(i) - what does the i stand for? Was that supposed to be the t or \alpha_t was supposed to be \alpha_i?

Minor stuff:
- Figure 5a is referred to as Table 5a in the text
- out-of-out-solver
- global graph embedding, figure 1 - G(phi, G), figure 2 - h(G)
- a figure of the policy architecture would be beneficial
- Figure 1
  - d_1 ->X OR Y in the graph is d1T, why isn’t it d1_OR, and connected to the OR node?
  - why isn’t d1_OR connected to OR node?
  - AST edge - but grammar is a DAG - (well, multigraph)
  - what are the reversed links? e.g. if A->B, reversed link is B->A ?
  - what is the meaning of the concrete figures in ‘one step’?
- consider relating to ‘DREAMCODER: Bootstrapping Domain-Specific Languages for Neurally-Guided Bayesian Program Learning’ (https://uclmr.github.io/nampi/extended_abstracts/ellis.pdf), as it’s another model that steps away from the fixed-DSL story

---

> ### Author Response · Authors · 2018-11-19
> **Response to Reviewer 2**
>
> We appreciate your effort in providing detailed and helpful reviews. We address the concerns and questions as follows:
>
> >> Cryptographic circuit synthesis tasks should consist of 214 tasks.
> We ignored 4 tasks that contain integer arithmetic operations (e.g. +), because circuit should only have logical operators.  To avoid confusion, we have now updated it to 214.
>
> >> How about other categories in SyGus competition?
> The other categories are not included in our evaluation due to two reasons. First, they have a very few number of tasks, most of which is around 30 or even fewer. Second, most tasks only have a few input/output example pairs, rather than a logical formal specification that is necessary for our approach to draw counterexamples.
>
> >> What is the setup difference from SyGus competition?
> The actual hardware and timeout limit are different. For each task, SyGus competition gives each solver 4-core 2.4GHz Intel processors with 128 GB memory and wallclock time limit of 1 hour. Our evaluation uses AMD Opteron 6220 processor, and assigns each solver a single core with 32 GB memory. We run each solver for 6 hours on each task. While our framework could take advantage of massively parallel hardware like GPUs, however, our evaluation does not use such hardware.
>
> >> Is ESymbolic a baseline?
>
> ESymbolic is a reasonable baseline because both ESymbolic and our framework use a top-down search based approach. ESymbolic expands a partial program by enumerating grammar rules in a fixed order, relies on the validity check of partially generated program by leveraging 2QBF (Quantified Boolean Formula), and backtracks immediately when the check fails.  However, our framework prioritizes grammar rules in the partial tree expansion based on the learned policy.
>
> >> Can you elaborate your choice for the state-of-the-art solver? EUSolver seems to the state-of-the-art.
> In terms of comparison with the state-of-the-art, we chose CVC4 solver over EUSolver, because CVC4 is a general SMT solver, whereas EUSolver is designed as a collection of specialized heuristics (e.g. indistinguishability and unification) for each benchmark category of the SyGuS competition, and (to our best knowledge) its design and implementation are guided and heavily tuned according to the SyGuS benchmarks. Our framework is also a general solver without requiring specialized heuristics for each domain. The speciality of EUSolver motivated us to develop a more general solver as baseline, namely ESymbolic, by replacing domain-specific heuristics used in EUSolver with a more general heuristic (i.e. partial program pruning with QBF).
>
> At the reviewer’s suggestion, we ran EUSolver with the same setup used in our evaluation. It solves 153 tasks (1 more task is solved in contrast with the SyGus 2017 report). These solved tasks are strictly a superset of those solved by CVC4 and ESymbolic. But EUSolver fails to solve 4 tasks solved by our framework. In terms of the absolute number of solved tasks, our framework is not yet as good as EUSolver, but it provides a new and complementary way to solve SyGuS tasks. We have incorporated this discussion in our revision.
>
> >> Can you describe how to calculate global graph embedding?
> Thanks for pointing this out. We simply sum over all the node embeddings to get the global graph embedding. We have clarified this in the revision.
>
> >> W for different edge types and different propagation steps t? Why is there a need for such a large number of parameters? What is the number of propagation steps?
> This is a general form of the Graph Neural Network. Since the #parameters is not the bottleneck in our task, we choose the most expressive parameterization. One could certainly choose to tie the weights in different layers. We use t=20 in all the experiments.

---

> > ### Author Response · Authors · 2018-11-19
> > **Response to Reviewer 2 (continue)**
> >
> > >> In the extreme case where all inputs can be enumerated - how often does this happen in the tasks you solve?
> > We randomly sample 100 inputs upfront for each task, which enumerates all inputs for 20 tasks with 6 (or less) variables, and a large fraction of inputs for 57 tasks with 7 variables. For the remaining tasks, we collect a new input (i.e. counter-example) and a few interpolated nearby inputs only when all current inputs have passed, which does not happen very often, and thus we do not end up enumerating all inputs for tasks with 8 or more variables.
> >
> > >> What is the meaning of \tau^(t-1) in figure-2? Is the tree on the right a generated subtree?
> > \tau^(t-1) is the partially generated program (\tau^(0) is the start symbol), which may contain non-terminals. The tree on the right shows the best rule that is going to be used to expand a particular non-terminal according to the current policy.
> >
> > >> Can you provide some intuition and details on state-tracking and state value estimator?
> > We use LSTM to track states throughout each episode starting from s0. S0 here is an embedding vector obtained from the graph embedding module that encodes the entire original program. For each RL step, we perform the following: (1) get the current state from LSTM; (2) use the current state to generate action and modify program tree; (3) use the embedding of the action to update LSTM; repeat until episode ends. When training using A2C, the error will back-prop end-to-end through both LSTM and graph embedder. The intuition to use LSTM to track the state is that we want the policy to be aware of its current context, i.e. how much progress on the tree has been made so far and this is reflected by the action taken so far. The value estimator is standard MLP with 128 nonlinear hidden units and linear outputs that takes the current state and outputs the estimated state value, which is used in A2C training.
> >
> >
> > >> the probability of each action (..) is defined as ….H_\alpha^(i) - what does the i stand for? Was that supposed to be the t or \alpha_t was supposed to be \alpha_i?
> > t and i are two different notions. \alpha_t here stands for the non-terminal node to be expanded at timestep t. For non-terminal node \alpha_t,  there are n_{\alpha_t} possible ways to expand.
> > For example, consider expand non-terminal s (s -> d1 OR d1 | d1 AND d1), then \alpha_t refers to s, and n_{\alpha_t} is 2. We define each of the expansions as the action and is associated with a embedding which is H_{\alpha_t}^{(i)}, so i here stands for the ith action among the n_{\alpha_t} possible ones.
> >
> > >> minor stuff
> > We apologize for certain unclear presentations and typos, which we have fixed in the revision.
> > For figure-1,   “d_1 -> X OR Y”  is meant to be “d_1 -> X | Y”. And yes, d1_OR should be connected to the global OR node. Each concrete sub-figure in “one step” shows a particular node sending/collecting messages to its neighbour nodes. Also, we will include DREAMCODER in our related work.

---

> > > ### Comment · AnonReviewer2 · 2018-11-25
> > > **Dissatisfied with the treatment of EUSolver in the original paper**
> > >
> > > > Difference from SyGus competition
> > >
> > > Could you be so kind to add the description of the difference to the paper? Appendix will do.
> > >
> > > > ESymbolic being a reasonable baseline (?)
> > >
> > > The question here was whether ESymbolic is considered a baseline or a SOTA engine, as the abstract mentions two SOTA engines, but I was not able to find ESymbolic as a SOTA model described anywhere else. Especially since its performance was way too low to be considered a SOTA engine. I now see the answers is no, and I see you corrected the wording in the text now.
> > >
> > > > EUSolver
> > >
> > > Albeit EUSolver performs better and significantly faster, I would not say its performance negatively impacts your contribution here. Especially as the argument of overspecialization vs generality would have been an easy one to digest. Would you please update your conclusion to reflect the relationship between the SOTAs and your model now, akin to how you updated the abstract?
> > >
> > > However, what I do not understand is why you did not include those results in the paper in the first place? Not just that, you did not even mention/cite the EUSolver in the original paper, as the obvious SOTA model (plus the the model is not correctly cited even right now). From the original paper, an uniformed reader would have easily been convinced that your model achieved SOTA on that subset of tasks, and that would have been the wrong conclusion to make. To reflect my dissatisfaction with this, I will lower my score to 6 for time being.

---

> > > > ### Author Response · Authors · 2018-11-26
> > > > **Thanks for helping us to correct the mistake**
> > > >
> > > > We apologize that the EUSolver was not evaluated in the first draft, which is in part due to the fact that we were not able to get the executable of EUSolver (and then collect results using our evaluation setup before the deadline). On the other hand, we thought CVC4 is the fairest representative as the state-of-the-art when designing experiments, as it is not specialized for a particular set of benchmarks.  But we agree with the reviewer that we should have made this very clear in the original paper. Now we are glad to see that we got the chance to finish the evaluation of EUSolver using our experiment setting and include its result in the revision.
> > > >
> > > > The conclusion is now updated, which was forgotten in revision 1 (sorry for that).
> > > >
> > > > EUSolver was actually cited in the related work of our original paper (see the reference: Rajeev Alur, Arjun Radhakrishna, and Abhishek Udupa, TACAS 2017). Now we also cite it in the experiment section and add the reference to SyGuS 2017 Competition evaluation report (arXiv:1711.11438).
> > > >
> > > >  We also add the description of our evaluation setup difference from SyGuS 2017 competition in the footnote of page 7, as it is fairly short.

---

> > > > > ### Comment · AnonReviewer2 · 2018-11-30
> > > > > **Removing dissatisfaction with the treatment of EUSolver in the original paper**
> > > > >
> > > > > I sympathize with your point of view, and consequently, I'm removing my previous dissatisfaction. Given the reiteration you did, you've got a good paper here.

---

> ### Comment · Area_Chair1 · 2018-11-30
> **Reassessment**
>
> Reviewer 2, thank you for participating in discussion with the authors. They appear to have been able to clarify some points that were of concern to you. Are you satisfied with this clarification? You seem to think the paper is good, but give it a borderline score. While we invite you to reconsider your assessment in light of the response, if you wish to stick by it, can you provide a short explanation as to why?

---

> > ### Comment · AnonReviewer2 · 2018-11-30
> > **Reassessment**
> >
> > My major concern was the treatment of EUSolver in the original paper, and that is why I lowered the score after the first iteration. Admittedly, the authors did address that issue, so I'm happy to go back to 7.

---

### Author Response · Authors · 2018-11-19
**Paper revision 1**

We updated our paper with the following changes:

- We fixed typos in Figure-1, improved Figure-2 and updated sec tion3.2 to clarify confusions about the graph representation.
- We added an evaluation of EUSolver at the reviewers' suggestion in section 5.
- We included a brief discussion about the recent DREAMCODER work in section 6.
- We also fixed a few other minor typos.

---

### Author Response · Authors · 2018-11-26
**Paper revision 2**

We updated our paper with the following changes:

- We updated the conclusion by making it consistent with the abstract updated in the previous revision.
- We included a few recent program synthesis work from ICML 2018 suggested by the anonymous reviewer into our references.

---

### Meta-Review · Area_Chair1 · 2018-12-13
**Exciting work**

**Confidence:** 4
**Recommendation:** Accept (Poster)

**Metareview:**

This paper presents an RL agent which progressively synthesis programs according to syntactic constraints, and can learn to solve problems with different DSLs, demonstrating some degree of transfer across program synthesis problems. Reviewers agreed that this was an exciting and important development in program synthesis and meta-learning (if that word still has any meaning to it), and were impressed with both the clarity of the paper and its evaluation. There were some concerns about missing baselines and benchmarks, some of which were resolved during the discussion period, although it would still be good to compare to out-of-the-box MCTS.

Overall, everyone agrees this is a strong paper and that it belongs in the conference, so I have no hesitation in recommending it.